# The Management of HIV Care Services in Central and Eastern Europe: Data from the Euroguidelines in Central and Eastern Europe Network Group

**DOI:** 10.3390/ijerph19137595

**Published:** 2022-06-21

**Authors:** Agata Skrzat-Klapaczyńska, Justyna D. Kowalska, Larisa Afonina, Svitlana Antonyak, Tatevik Balayan, Josip Begovac, Dominik Bursa, Gordana Dragovic, Deniz Gokengin, Arjan Harxhi, David Jilich, Kerstin Kase, Botond Lakatos, Mariana Mardarescu, Raimonda Matulionyte, Cristiana Oprea, Aleksandr Panteleev, Antonios Papadopoulos, Lubomir Sojak, Janez Tomazic, Anna Vassilenko, Marta Vasylyev, Antonija Verhaz, Nina Yancheva, Oleg Yurin, Andrzej Horban

**Affiliations:** 1Department of Adults’ Infectious Diseases, Hospital for Infectious Diseases, Medical University of Warsaw, 02-091 Warszawa, Poland; jdkowalska@gmail.com (J.D.K.); dominikbursa@wp.pl (D.B.); ahorban@zakazny.pl (A.H.); 2Republic Clinical Hospital for Infections of MoH of the Russian Federation, 101000 Moscow, Russia; alarissa-ridh@rambler.ru; 3Viral Hepatitis and AIDS Department, Gromashevsky Institute of Epidemiology and Infectious Diseases, 01001 Kyiv, Ukraine; antonyaksn@gmail.com; 4National Center for Disease Control and Prevention, Yerevan 0002, Armenia; tatevikbalayan@gmail.com; 5School of Medicine, University Hospital for Infectious Diseases, University of Zagreb, 10000 Zagreb, Croatia; josip.begovac@gmail.com; 6Department of Pharmacology, Clinical Pharmacology and Toxicology, School of Medicine, University of Belgrade, 11000 Belgrade, Serbia; gozza@beotel.net; 7Department of Infectious Diseases and Clinical Microbiology, Faculty of Medicine, Ege University, 35040 Izmir, Turkey; gkengin61@gmail.com; 8Infectious Disease Service, University Hospital Center of Tirana, 1001 Tirana, Albania; harxhiarjan@yahoo.com; 9Department of Infectious Diseases, 1st Faculty of Medicine, Charles University in Prague and Faculty Hospital Bulovka Hospital, 18000 Prague, Czech Republic; david.jilich@centrum.cz; 10West Tallinn Central Hospital, 10111 Tallinn, Estonia; doktorkase@gmail.com; 11National Institute of Hematology and Infectious Diseases, South-Pest Central Hospital, National Center of HIV, 1007 Budapest, Hungary; btlakatos@gmail.com; 12National Institute for Infectious Diseases Matei Bals Bucharest, 021105 Bucharest, Romania; mardarescum@yahoo.com; 13Faculty of Medicine, Vilnius University, Vilnius University Hospital Santaros Klinikos, 08410 Vilnius, Lithuania; raimonda.matulionyte@santa.lt; 14Victor Babes Clinical Hospital for Infectious and Tropical Diseases, Carol Davila University of Medicine and Pharmacy, 010001 Bucharest, Romania; cristiana.oprea@spitalulbabes.ro; 15City TB Dispensary, 101000 Moscow, Russia; alpanteleev@gmail.com; 16University General Hospital Attikon, Medical School, National and Kapodistrian University of Athens, 15772 Athens, Greece; antpapa1@otenet.gr; 17Department of Infectology and Geographical Medicine, Center for Treatment of HIV/AIDS Patients, Academic L. Derer’s University Hospital, 2412 Bratislava, Slovakia; lubosojak@centrum.sk; 18Clinic for Infectious Diseases, University Medical Centre Ljubljana, 1000 Ljubljana, Slovenia; janez.tomazic@kclj.si; 19Global Fund Grant Management Department, Republican Scientific and Practical Center for Medical Technologies, 220004 Minsk, Belarus; anne.vassilenko@gmail.com; 20Astar Medical Center, 79007 Lviv, Ukraine; vasylyevmarta@gmail.com; 21Department for Infectious Diseases, Faculty of Medicine, University of Banja Luka, 78000 Banja Luka, Republika Srpska, Bosnia and Herzegovina; antonija@blic.net; 22Department for AIDS, Specialized Hospital for Active Treatment of Infectious and Parasitic Disease Sofia, 1000 Sofia, Bulgaria; nyancheva@gmail.com; 23Central Research Institute of Epidemiology, Federal AIDS Centre, 101000 Moscow, Russia; oleg_gerald@mail.ru

**Keywords:** HIV, COVID-19, models of care, Central and Eastern Europe, telehealth

## Abstract

Introduction: The COVID-19 pandemic has been challenging time for medical care, especially in the field of infectious diseases (ID), but it has also provided an opportunity to introduce new solutions in HIV management. Here, we investigated the changes in HIV service provision across Central and Eastern European (CEE) countries before and after the COVID-19 outbreak. Methods: The Euroguidelines in Central and Eastern Europe Network Group consists of experts in the field of ID from 24 countries within the CEE region. Between 11 September and 29 September 2021, the group produced an on-line survey, consisting of 32 questions on models of care among HIV clinics before and after the SARS-CoV-2 outbreak. Results: Twenty-three HIV centers from 19 countries (79.2% of all countries invited) participated in the survey. In 69.5% of the countries, there were more than four HIV centers, in three countries there were four centers (21%), and in four countries there was only one HIV center in each country. HIV care was based in ID hospitals plus out-patient clinics (52%), was centralized in big cities (52%), and was publicly financed (96%). Integrated services were available in 21 clinics (91%) with access to specialists other than ID, including psychologists in 71.5% of the centers, psychiatrists in 43%, gynecologists in 47.5%, dermatologists in 52.5%, and social workers in 62% of all clinics. Patient-centered care was provided in 17 centers (74%), allowing consultations and tests to be planned for the same day. Telehealth tools were used in 11 centers (47%) before the COVID-19 pandemic outbreak, and in 18 (78%) after (*p* = 0.36), but were represented mostly by consultations over the telephone or via e-mail. After the COVID-19 outbreak, telehealth was introduced as a new medical tool in nine centers (39%). In five centers (28%), no new services or tools were introduced. Conclusions: As a consequence of the COVID-19 pandemic, tools such as telehealth have become popularized in CEE countries, challenging the traditional approach to HIV care. These implications need to be further evaluated in order to ascertain the best adaptations, especially for HIV medicine.

## 1. Introduction

COVID-19 has been a challenging time for all medical services, especially at the beginning of the pandemic, and it has been difficult to maintain continued care for HIV patients [1]. After almost two years, it is still a challenge, but it is also an opportunity for Central and Eastern Europe (CEE) to modernize old systems and introduce modern tools to HIV care [2].

The CEE region is characterized by large disproportions in quality and access to medical care for HIV patients [3,4]. Moreover, the epidemiological diversity in the representation of key populations, such as injection drug users (IDU) or men that have sex with men (MSM), makes it difficult to adapt the same model of care everywhere [5]. What remains replicable across the CEE region is an approach that provides an integrated and patient-centered model of care, based on in-person visits. However, due to the sudden necessity for social distancing and lock downs, such models have not been able to continue. Healthcare workers (HCWs) doing their job under the pressure of the pandemic, were exposed to stress, depression and anxiety [6]. It also had an impact on the quality of medical services provided and HCWs’ psychological well-being and job performance [7]. In addition, during the COVID-19 crisis, infectious diseases (ID) workforces were seriously overloaded and resources were shredded, making it impossible to maintain previous models of medical care delivery for HIV-infected patients [8]. Such obstacles have been proven to influence the spectrum of testing and prevention services, but their impact on cART delivery, and thus viral suppression, is not clear [9]. A barrier to effective patient care, in areas other than ID, could be the stigmatizing attitude towards people living with HIV (PLWH) [10,11]. This situation causes difficulties in providing health care to patients in this population. So far, there has been no evidence of major disruption in the continuum of HIV care, which may contribute to a change in the services adapting to the new COVID-19 pandemic situation [12].

However, as mentioned above, the provision of HIV care, which requires strict adherence to antiretroviral regimens and careful follow up to ascertain viral suppression, has been built on several principles, most of which require in-person care delivery and support. In general, the integrated model of HIV care is based on the use of a wide range of primary and specialized medical services, which allow HIV patients to be treated in a multidisciplinary manner [13]. On the other hand, the patient-centered HIV model of care is based on the strong relationship and collaboration between care providers, pharmacists, and patients [14].

Here we investigate to what extent the COVID-19 pandemic has initiated changes in HIV care delivery in HIV care centers in the CEE.

## 2. Methods

The Euroguidelines in Central and Eastern Europe (ECEE) Network Group was established in February 2016 to promote standards of care for HIV and viral hepatitis infections in the region. The group includes experts in the field of infectious diseases from 24 countries in the region, who are also professionals actively involved in the care of infectious diseases [15]. At the beginning of September 2021, the group produced an on-line questionnaire, based on the SurveyMonkey^®^ platform, consisting of 32 questions (see Appendix A). The survey focused on three main areas, namely, models of HIV care, the availability of HIV medical care, and new tools introduced into HIV health services during the COVID-19 outbreak. Respondents were recruited from ECEE members and contacted by e-mail. The inclusion criteria for the study consisted of being actively involved in HIV medical care and employment in the HIV medical center. The survey was prepared in English and the definition of telehealth was provided in the questionnaire. Data were collected from Albania, Armenia, Belarus, Bosnia and Herzegovina, Bulgaria, Croatia, Czech Republic, Estonia, Greece, Hungary, Lithuania, Poland, Romania, Russia, Serbia, Slovakia, Slovenia, Turkey and Ukraine. The data were collected until the end of September 2021.

Telehealth was defined as the delivery of health care, health education, and health information services via remote technologies [16].

In the statistical analyses, McNemar’s test was used for group comparison as appropriate.

The study was approved by the Bioethical Committee of the Medical University of Warsaw (Nr AKBE/219/2021).

## 3. Results

Twenty-three respondents representing 23 HIV medical centers from 19 countries participated in the survey (83% response rate). The majority of respondents (20, 87%) were infectious diseases specialists. Three (13%) of the respondents were not ID physicians (other specialty physician, other medical personnel, other non-medical personnel). Fourteen (60%) respondents were women. In 12 cases (63%), there were more than four HIV medical care centers in each country, in three countries there were four centers (16%), and in four countries (Albania, Slovenia, Bosnia and Herzegovina, and Armenia; 21%) there was only one HIV center. HIV medical care was mostly based in ID hospitals plus out-patient clinics (12, 52%), was centralized in big cities (12, 52%), and was publicly financed in 22 centers (95.5%). The average number of patients visiting HIV medical centers ranged between 100 and 3500 per year.

In terms of linkage to care, HIV testing facilities were included in HIV medical centers in 17 cases (74%). It was not possible to register remotely for a visit in only two centers (9%). In 22 (95.5%) HIV centers, there were no waiting lists for the first visit, and in 23 centers (100%), there were no waiting lists for ART.

### 3.1. Integrated Services

Integrated services were available in 21 clinics (91%) with access to specialists other than ID, including psychologists in 15 centers (71.5%), psychiatrists in 8 (38%), gynecologists in 11 (52.5%), dermatologists in 12 (57%), and social workers in 14 clinics (66%).

Integrated medical care for HIV-positive women in the centers where gynecologists were available also covered a few other areas, especially cancer screening in 43.5% of the centers, contraception in 43.5%, and pregnancy in 39%.

### 3.2. Patient-Centered Care

Seventeen centers (74%) provided patient-centered care, allowing all necessary consultations and tests to be planned on the same day. Moreover, it was also possible to diagnose and treat sexually transmitted infections (STIs) in 17 centers (74%). The most commonly treated STI was syphilis, in 20 centers (87%); then chlamydia, in 17 centers (74%); and gonorrhea, in 16 centers (69.5%). The availability of HPV treatment was reported by 11 centers (48%).

In addition, in 21 centers (91%) it was a possible to consult the HIV-negative partners of HIV-positive patients. Fifteen clinics (65%) offered testing for HIV, HBV, HCV, and other sexually transmitted infections. Besides this, clinics also offered STI treatment for HIV-negative partners in 14 (61%) of the centers; they could also receive a PrEP prescription in 12 (52%) of the HIV medical centers.

In CEE countries, there was less access for specific groups of patients in 10 centers (43.5%); these were mostly prisoners in seven centers (30.5%), migrants in five (22%), and IVDU in four (17%).

### 3.3. Telehealth and New Tools

Telehealth tools were used in 11 centers (47%) before, and in 18 centers (78%) after the COVID-19 pandemic outbreak (*p* = 0.36) (Figure 1) but were represented mostly by consultations over the telephone or via e-mail (Figure 2). After the COVID-19 outbreak, telehealth was introduced in nine centers (39%) as a new medical tool.

In addition, in a qualitative assessment, new services were introduced including home-based HIV testing (3, 13%), longer periods between the timing of drug supplies (11, 48%), medication delivery programs (7, 30%), community ART distribution (1, 4.5%), home visits (2, 9%), and mobile health centers (3, 13%) were created (Figure 3). In five centers (21%), no new services or tools had been introduced by September 2021.

## 4. Discussion

Our study indicates that HIV medical care in CEE countries before the COVID-19 pandemic was based on an integrated model of care, and to a lesser extent, on patient-centered care. Although the most important goal of medical care is to achieve viral suppression, the complexity of HIV care, as well as the number of comorbidities and associated social problems requires the creation of a wide range of medical services [17]. This situation has resulted in the introduction of a model of care whereby the primary service is supported at the point of care by integrating non-infectious diseases providers such as psychiatrics, cardiologists, and social workers [18]. The patient’s well-being is the result of high-level medical services, but also the integration of these services, as in the case of the health of HIV-infected women. In our study, integrated medical care for HIV-positive women in the centers where gynecologists were available also covered several other areas, especially cancer screening, contraception and pregnancy, but also family planning, menopause, and STI screening for women. However, over the years it has been recognized that individualized HIV medical care that respects and responds to patients’ preferences—that is, patient-centered care—is relevant to patients with complex issues and many problems, especially HIV-infected patients [19]. What should be taken into account, is that people living with HIV are aging and their health needs are becoming greater, which, in the case of the underlying disease, creates the need for patient-centered care [20].

The progress in technological developments, which have been systematically implemented in health systems, means that there is now a need to remodel the current HIV medical systems [21]. The COVID-19 pandemic has offered us the opportunity for a new era of HIV management by increasing the availability of HIV care services through the introduction of telehealth and the creation of additional care and treatment options.

During the study period, a significant increase in COVID-19 cases was observed worldwide. Despite the overloaded health care systems, the solutions introduced to maintain continuity of care for HIV-infected patients may make the current system much more efficient outside the COVID-19 pandemic outbreak. In a study by Kowalska et al., at the end of March 2020, only 31.6% of the HIV clinics in Central and Eastern European countries were working normally; 57.9% of clinicians were involved in HIV care and also in the care of patients diagnosed with COVID-19; moreover, 52.6% of HIV patients were under COVID-19 quarantine. HIV health systems were overloaded and in 42.1% of cases, HIV drug procurement might have been affected [8]. Additionally, Western Europe reported that there were significant losses in the 90–90–90 objectives in 2020, which were associated with the introduction of lockdowns and limited access to HIV outpatient services [1]. This situation, worldwide, led to new ideas and solutions that allowed the continuity of care for HIV patients to be maintained during this difficult time for the HIV epidemic. In Central and Eastern European countries, telehealth tools were used by less than half the responding centers, and were represented by a modest choice of phone and/or e-mail communication. The current state of telehealth provides a much wider choice of tools, including video consultations, and even video conferences with additional providers such as social workers or personal health assistants [22]. The broad introduction of telemedicine could help to compensate for the reduction in face-to-face patient–physician encounters, which was always at the the core of HIV care, and was extremely useful during the pandemic [23]. On the positive side, this is the first time that telemedicine tools have been introduced in some CEE countries, and in total, their use increased by almost 80%. Before the pandemic their use was limited to narrow and specialized populations such as cardiology, psychiatry, and ophthalmology [24,25]. Telehealth has not been evaluated for use in HIV care in the CEE region, where specific challenges exist (e.g., less access to teletechnology). Although challenging, telehealth could be an opportunity to accelerate communication with patients from rural areas who are unable to travel due to work, family duties, or disability. While face-to-face patient–physician meetings are essential for HIV medical care, especially at the beginning of the diagnosis process, this could be combined with telehealth later in the care process [26,27]. Although some studies show that personal contact with patients significantly improves visit adherence and promotes better relationships with the therapeutic team in primary care, they do not compare this contact with specific telehealth tools as an alternative option [28,29]. In addition, studies from the HIV field in this area are scarce. Researchers have proven, however, that canceled visits are associated with higher rates of non-suppression of viral load [30]. Although reducing patient–doctor visits in real life may result in an increase in long-term mortality, it has not been verified whether telehealth is capable of filling this important gap [31].

For many patients, telehealth is a very convenient, modern form of contact with their treatment center and it does not interfere with their professional or personal life [32]. However, it is possible that some groups of patients do not have the same access or understanding of the tools. Whether or not telehealth is jeopardizing access to care across different subgroups of HIV-positive patients remains an important question yet to be answered by research. The widespread introduction of telehealth will have an impact on patient outcomes. The convenient appointment times, reduction in stigma-related delays in care or broader access to care for those living in rural areas who have problems due to the lack of transport may have a positive impact on the patient’s satisfaction. However, the use of telehealth in HIV care also has several key challenges, such as access to technological resources or loss of face-to-face personal interactions, that may have a negative effect on patient outcomes [33].

There are some limitations to our study that should be mentioned. First of all, the introduction of telehealth and other tools in medical practice during the COVID-19 pandemic is a dynamic process and our study, which has a cross-sectional nature, is not able to capture this. Second, we did not include all aspects of HIV care in the questionnaire. This study represents the perceived opinion of providers, and moreover, as not all the centers in a given country responded to the questionnaire, the answers may not be representative of those countries. Moreover, the survey was taken from the service-side instead of the recipient-side. Therefore, it is unclear as to how the service-users felt the impacts of COVID-19 on the service delivery. In addition, the survey was self-reported and it is possible that it is biased in some aspects.

## 5. Conclusions

In conclusion, due to the COVID-19 outbreak, most audited HIV centers in the CEE introduced telehealth services. This is a challenge, but also an opportunity for a new era of HIV management, and possibly, a chance to revitalize healthcare infrastructure tailored to patients’ needs. The combination of traditional patient-centered, face-to-face care, and the provision of telehealth and other new tools in HIV medical care may be beneficial but should be further investigated. The COVID-19 pandemic has created an opportunity for such research.

## Figures and Tables

**Figure 1 ijerph-19-07595-f001:**
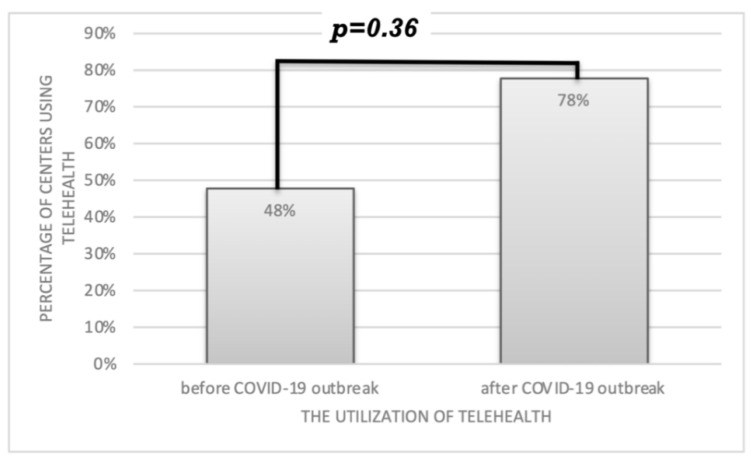
The utilization of telehealth before and after the COVID-19 outbreak in 23 HIV centers in Eastern and Central Europe (*p*-value 0.36).

**Figure 2 ijerph-19-07595-f002:**
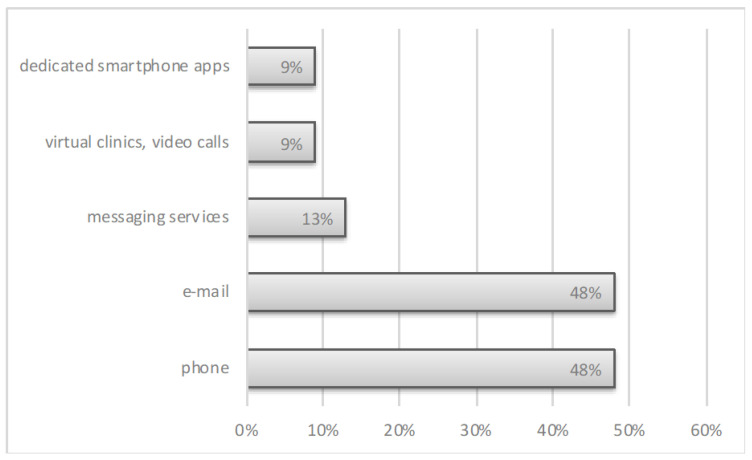
The types of telehealth used in HIV medical care before the COVID-19 outbreak.

**Figure 3 ijerph-19-07595-f003:**
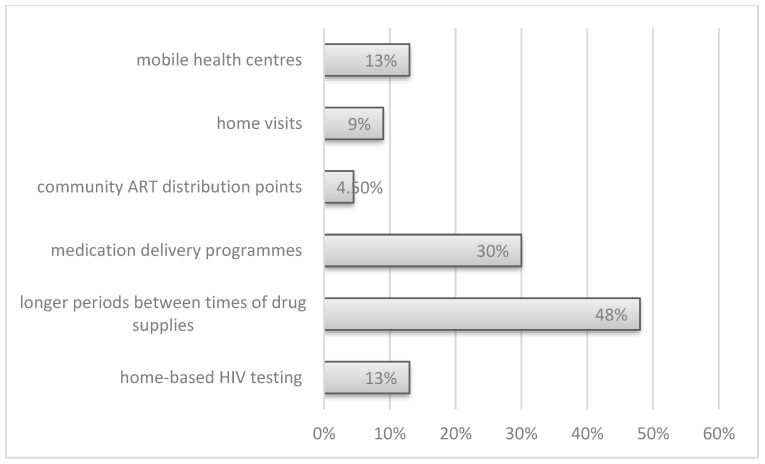
The new tools introduced into HIV medical care after the COVID-19 outbreak.

## Data Availability

The data presented in this study are available from the corresponding author on reasonable request.

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
