# Peer review of "The Management of HIV Care Services in Central and Eastern Europe: Data from the Euroguidelines in Central and Eastern Europe Network Group"

_ijerph, 2022, doi:10.3390/ijerph19137595_

Round 1
Reviewer 1 Report
This manuscript is very specific to a HIV care and Central and Eastern Europe and is insightful how care is being managed during the pandemic. Below are some specific comments:
1. The title is a little misleading in that it is not necessarily reporting the impact of pandemic on medical care because there are no pre- and post-pandemic comparisons with the exception of Telehealth. Perhaps the authors should consider changing the title to reflect the management of HIV care rather than the impact of the pandemic.
2. The authors put a lot of emphasis on Telehealth, which has been more widely utilized during the pandemic. The incorporation of Telehealth occurs across all areas of medical care and not just in HIV. Thus, it would be useful for the authors to provide some insight on how Telehealth has specifically impacted HIV care and whether it has had a positive or negative effect on patient outcomes.
3. Also, the findings were based on the opinions of the providers which may differ from those of the patients. In continuation of the above comment, future aims could incorporate the opinions of the patients to determine how these changes in HIV care have impacted them directly.
Author Response
Reviewer 1
This manuscript is very specific to a HIV care and Central and Eastern Europe and is insightful how care is being managed during the pandemic. Below are some specific comments:
1.The title is a little misleading in that it is not necessarily reporting the impact of pandemic on medical care because there are no pre- and post-pandemic comparisons with the exception of Telehealth. Perhaps the authors should consider changing the title to reflect the management of HIV care rather than the impact of the pandemic.
This has been changed. The title is now: “The Management of HIV Care Services in Central and Eastern Europe: Data from the Euroguidelines in Central and Eastern Europe Network Group”
2. The authors put a lot of emphasis on Telehealth, which has been more widely utilized during the pandemic. The incorporation of Telehealth occurs across all areas of medical care and not just in HIV. Thus, it would be useful for the authors to provide some insight on how Telehealth has specifically impacted HIV care and whether it has had a positive or negative effect on patient outcomes.
This has been added in the Discussion section and the new reference was added.
3. Also, the findings were based on the opinions of the providers which may differ from those of the patients. In continuation of the above comment, future aims could incorporate the opinions of the patients to determine how these changes in HIV care have impacted them directly.
This aspect will be analyzed in the future studies. Thank you for that comment

Reviewer 2 Report
I read this submission with interests because it is important to know the health service delivery during the COVID-19 pandemic, especially for those vulnerable populations (i.e., people with HIV in the present submission). However, there are some issues in the present submission. Please see my comments below.
1. In the first paragraph, when describing the impacts of COVID-19, please elaborate more on the general impacts before going to the specific population of those with HIV. Please see the following references for support.
Rajabimajd N, Alimoradi Z, Griffiths MD. Impact of COVID-19-related fear and anxiety on job attributes: A systematic review. Asian J Soc Health Behav 2021;4:51-5
Prasiska DI, MUHLIS AN, Megatsari H. Effectiveness of the emergency public activity restrictions on COVID-19 epidemiological parameter in East Java Province, Indonesia: An ecological study. Asian J Soc Health Behav 2022;5:33-9
Sangma RD, Kumar P, Nerli LM, Khanna AM, Vasavada DA, Tiwari DS. Social stigma and discrimination in Coronavirus Disease-2019 survivors and its changing trend: A longitudinal study at tertiary care center Gujarat, India. Asian J Soc Health Behav 2022;5:68-74
Shirali GA, Rahimi Z, Araban M, Mohammadi MJ, Cheraghian B. Social-distancing compliance among pedestrians in Ahvaz, South-West Iran during the Covid-19 pandemic. Asian J Soc Health Behav 2021;4:131-6
Olashore AA, Akanni OO, Fela-Thomas AL, Khutsafalo K. The psychological impact of COVID-19 on health-care workers in African Countries: A systematic review. Asian J Soc Health Behav 2021;4:85-97
2. In the Introduction, it will be good to mention the difficulties of providing healthcare for people with HIV as they have the issues of stigma. Please see the following special issue in the IJERPH. I believe that there are some good papers on people with HIV to help the authors to improve their Introduction.
https://www.mdpi.com/journal/ijerph/special_issues/stigma_2
3. The Methods should be improved with providing the following information: (i) who were those who represent each HIV service to complete the survey (i.e., what were the eligibility, inclusion criteria, and exclusion criteria for the participants); (ii) what language was used for the survey (if there were local language used, how was the translation conducted); (iii) was the definition of telehealth clearly introduced to the respondents.
4. The data analysis seems inadequate. Specifically, the comparison of using telehealth before and after the COVID-19 outbreak should not be tested using chi-square test. Please consult a statistician for proper statistics used for the present study.
5. The Results should be improved with providing the following information: (i) the demographic information of the respondents (e.g., position in the HIV service, age, sex); (ii) the coverage size of each HIV service (i.e., how many people with HIV received the HIV service).
6. I think that the authors should acknowledge other limitations, including (i) the survey was taken apart from the service-side instead of the recipient-side. Therefore, it is unclear how the service-users felt the impacts of COVID-19 on the service delivery; (ii) the survey was self-reports and it is possible to be biased by the social desirability.
7. The authors mentioned that they have a supplementary to describe the survey questions. However, I did not see any supplementary in the system.
Author Response
Reviewer 2
I read this submission with interests because it is important to know the health service delivery during the COVID-19 pandemic, especially for those vulnerable populations (i.e., people with HIV in the present submission). However, there are some issues in the present submission. Please see my comments below.
- In the first paragraph, when describing the impacts of COVID-19, please elaborate more on the general impacts before going to the specific population of those with HIV. Please see the following references for support.
Rajabimajd N, Alimoradi Z, Griffiths MD. Impact of COVID-19-related fear and anxiety on job attributes: A systematic review. Asian J Soc Health Behav 2021;4:51-5
Prasiska DI, MUHLIS AN, Megatsari H. Effectiveness of the emergency public activity restrictions on COVID-19 epidemiological parameter in East Java Province, Indonesia: An ecological study. Asian J Soc Health Behav 2022;5:33-9
Sangma RD, Kumar P, Nerli LM, Khanna AM, Vasavada DA, Tiwari DS. Social stigma and discrimination in Coronavirus Disease-2019 survivors and its changing trend: A longitudinal study at tertiary care center Gujarat, India. Asian J Soc Health Behav 2022;5:68-74
Shirali GA, Rahimi Z, Araban M, Mohammadi MJ, Cheraghian B. Social-distancing compliance among pedestrians in Ahvaz, South-West Iran during the Covid-19 pandemic. Asian J Soc Health Behav 2021;4:131-6
Olashore AA, Akanni OO, Fela-Thomas AL, Khutsafalo K. The psychological impact of COVID-19 on health-care workers in African Countries: A systematic review. Asian J Soc Health Behav 2021;4:85-97
This has been added in the Intruduction section. Two references:
Rajabimajd N, Alimoradi Z, Griffiths MD. Impact of COVID-19-related fear and anxiety on job attributes: A systematic review. Asian J Soc Health Behav 2021;4:51-5
Olashore AA, Akanni OO, Fela-Thomas AL, Khutsafalo K. The psychological impact of COVID-19 on health-care workers in African Countries: A systematic review. Asian J Soc Health Behav 2021;4:85-97
were also added
- In the Introduction, it will be good to mention the difficulties of providing healthcare for people with HIV as they have the issues of stigma. Please see the following special issue in the IJERPH. I believe that there are some good papers on people with HIV to help the authors to improve their Introduction.
https://www.mdpi.com/journal/ijerph/special_issues/stigma_2
This has been mentioned and the references from the special issue were added
- The Methods should be improved with providing the following information: (i) who were those who represent each HIV service to complete the survey (i.e., what were the eligibility, inclusion criteria, and exclusion criteria for the participants); (ii) what language was used for the survey (if there were local language used, how was the translation conducted); (iii) was the definition of telehealth clearly introduced to the respondents.
This has been added in the Methods section. Regarding translation from English to local languages. It was not necessary. All ECEE Network Group Members speak English fluently.
- The data analysis seems inadequate. Specifically, the comparison of using telehealth before and after the COVID-19 outbreak should not be tested using chi-square test. Please consult a statistician for proper statistics used for the present study.
Is has been consulted with the statistician, the non-parametric test (Chi2 test) was used for group comparison as appropriate.
- The Results should be improved with providing the following information: (i) the demographic information of the respondents (e.g., position in the HIV service, age, sex); (ii) the coverage size of each HIV service (i.e., how many people with HIV received the HIV service).
This has been added in the Results section.
- I think that the authors should acknowledge other limitations, including (i) the survey was taken apart from the service-side instead of the recipient-side. Therefore, it is unclear how the service-users felt the impacts of COVID-19 on the service delivery; (ii) the survey was self-reports and it is possible to be biased by the social desirability.
This has been added to the Discussion section, limitations paragraph.
- The authors mentioned that they have a supplementary to describe the survey questions. However, I did not see any supplementary in the system.
Please see below, we will try to add the supplementary materials add again to the system
Questionnaire – models of HIV care during COVID-19 times
The achievement of global 90-90-90 targets in HIV testing, prevention and treatment was severely threatened by the COVID-19 pandemic. In this context, there was a need for new models of HIV care.
In this questionnaire we are asking what new strategies of HIV care were implemented during challenging COVID-19 time in Central and Eastern Europe Countries and answers for the relevant questions – how we improved service for key populations and how we can adapt the lessons learned to the post-COVID-19 period.
Telehealth - the provision of healthcare remotely by means of telecommunications technology
PERSONAL DATA
- Full name and surname:…………………….
- E-mail………..
- Country:………………..
- Clinic full name (affiliation):………………………………..
- Medical speciality: ………………………….
- ID physician (completed or in curse)
- Other specialty physician
- Other medical personnel
- Other non-medical personnel
GENERAL INFORMATION
- Please describe the organisation of your centre:
- based on out-patient clinics
- ID hospital plus out-patients clinic
- general hospital plus out-patient clinic
- other (please specify)
- Are you directly involved in HIV care:
- Yes
- No
- Other:………………………….
- Is your country in a lock-down situation at the moment?
- Yes
- No
- No, but a lock-down is likely soon
- No, we have completed lock-down
- Was your center affected by the COVID-19 outbreak?
- Yes
- no
- How is your HIV clinic operating now:
- Normally
- Shorter hours but admitting
- Less personnel but admitting
- Normal activity suspended
- HIV clinic is closed
- We combine face-to face visits and telehealth
- Other: …………………………
TESTING
- Are people in your country able to access testing for HIV?
- Yes
- No
- Was there any decline in testing for HIV after March 2020?
- Yes
- No
- I don’t know
- If yes, what were the reasons? (multiple answer possible)
- testing facilities were closed
- people were unable to travel to health facilities due to travel restrictions
- people were avoiding going to testing facilities due to COVID-19
- the lack of staff
- other
- Were there any new testing for HIV methods introduced after March 2020?
- Rapid self tests
- Mobile testing points
- Other (please specify)
- No
NEW MODELS OF CARE
- Were there any telehealth tools used before COVID-19 pandemic as a routine?
- Yes
- No
- If yes, what telehealth tools were used before COVID-19 pandemic as a routine in your center?
- phone
- other messaging services
- virtual clinics, videocalls (eg. via Skype)
- on-line health parameters assessment (eg. ECG)
- dedicated smartphone apps
- other (please specify)
- Which new services or tools (if any) were introduced after March 2020?
- Telehealth (the provision of healthcare remotely by means of telecommunications technology.)
- home-based HIV testing
- longer period of time in drug supplies
- Medication delivery programmes
- Community ART distribution points
- Home visits
- Mobile health centres
- None of the above
- Other (please specify)
- For how long were you supplying ARVs for your patients before March 2020:
- 2 weeks
- 1 month
- 2-3 months
- 4-6 months
- For how long were you supplying ARVs for your patients after March 2020 by May 2021?
- 2 weeks
- 1 month
- 2-3 months
- 4-6 months
- More
- For how long are you supplying ARVs for your patients now?
- 2 weeks
- 1 month
- 2-3 months
- 4-6 months
- More
- How often had you measured the viral load in your patients before March 2020?
- Every 1 month
- Every 3 months
- Every 6 months
- Other
- How often had you measured the viral load in your patients after March 2020 by May 2021?
- Every 1 month
- Every 3 months
- Every 6 months
- Other
- How often do you measure the viral load in your patients now?
- Every 1 month
- Every 3 months
- Every 6 months
- Other
- How much did face-to face visits declined after March 2020 by the end of 2020 (in compare to the times before COVID-19 pandemic)
- 0%,
- 0-25%,
- 25-50%
- >50%
- How much did face-to face visits declined from the end of 2021 by the end of May 2021 (in compare to the times before COVID-19 pandemic)
- 0%,
- 0-25%,
- 25-50%
- >50%
KEY POPULATIONS
- Did you have lower access to specific group of patients during COVID-19 pandemic:
- Yes
- No
- If yes, which group it was related to:
- Migrants
- IVDU
- Chemsex users
- People in prisons
- Other (please specify)
- If yes, what was the probable reason?related to current pandemic situation (eg. Social worker or translator not available, mobile clinics not operating): …………….
- Was there any tools in your center to take care about above patients?
- Please make a comment
- Any other comments (up to 50 words): ………………………………………..

Round 2
Reviewer 2 Report
The authors have replied to my previous comments satisfactorily, except for the use of chi-square test. It is not about the parametric or non-parametric in the test. It is about that the two compared values were dependent or independent. Chi-square test assumes the two compared values should be independent (e.g., comparing the service use percentages between male and female participants). However, if the authors aimed to compare telehealth use rates before and during the COVID-19 pandemic from the same sample , the two values are dependent. In this case, chi-square test is inadequate. McNemar test should be the choice. However, please correct me if the two values were not dependent (but the reading in the manuscript implies that they are dependent).
Author Response
The authors have replied to my previous comments satisfactorily, except for the use of chi-square test. It is not about the parametric or non-parametric in the test. It is about that the two compared values were dependent or independent. Chi-square test assumes the two compared values should be independent (e.g., comparing the service use percentages between male and female participants). However, if the authors aimed to compare telehealth use rates before and during the COVID-19 pandemic from the same sample , the two values are dependent. In this case, chi-square test is inadequate. McNemar test should be the choice. However, please correct me if the two values were not dependent (but the reading in the manuscript implies that they are dependent).
It has been corrected in the Methods and also on the Results section - we've chosen McNemar test.